# Long-Term Impact of COVID-19 on Hospital Visits of Rural Residents in Guangdong, China: A Controlled Interrupted Time Series Study

**DOI:** 10.3390/ijerph192013259

**Published:** 2022-10-14

**Authors:** Wenfang Zhong, Rong Yin, Yan Pan, Xiangliang Zhang, Andre M. N. Renzaho, Li Ling, Xingge Li, Wen Chen

**Affiliations:** 1Department of Medical Statistics, School of Public Health, Sun Yat-sen University, Guangzhou 510080, China; 2Translational Health Research Institute, School of Medicine, Western Sydney University, Campbelltown 2560, Australia; 3Maternal, Child and Adolescent Health Program, Burnet Institute, Melbourne 3004, Australia; 4Sun Yat-sen Center for Migrant Health Policy, Sun Yat-sen University, Guangzhou 510080, China

**Keywords:** COVID-19, impact, healthcare-seeking behavior, ICD-10, interrupted time series analysis

## Abstract

To date, there is a lack of comprehensive understanding regarding the effect of coronavirus disease 2019 (COVID-19) on the healthcare-seeking behavior and utilization of health services in rural areas where healthcare resources are scarce. We aimed to quantify the long-term impact of COVID-19 on hospital visits of rural residents in China. We collected data on the hospitalization of all residents covered by national health insurance schemes in a county in southern China from April 2017 to March 2021. We analyzed changes in residents’ hospitalization visits in different areas, i.e., within-county, out-of-county but within-city, and out-of-city, via a controlled interrupted time series approach. Subgroup analyses based on gender, age, hospital levels, and ICD-10 classifications for hospital visits were examined. After experiencing a significant decline in hospitalization cases after the COVID-19 outbreak in early 2020, the pattern of rural residents’ hospitalization utilization differed markedly by disease classification. Notably, we found that the overall demand for hospitalization utilization of mental and neurological illness among rural residents in China has been suppressed during the pandemic, while the utilization of inpatient services for other common chronic diseases was redistributed across regions. Our findings suggest that in resource-poor areas, focused strategies are urgently needed to ensure that people have access to adequate healthcare services, particularly mental and neurological healthcare, during the COVID-19 pandemic.

## 1. Introduction

The coronavirus disease 2019 (COVID-19) pandemic is changing long-established healthcare-seeking behaviors around the world [1,2,3,4,5]. Due to the fear of contracting COVID-19 [1], the booming business of telemedicine [6], the relatively complicated procedures of seeking healthcare services during the COVID-19 pandemic [7], and strict government restrictions on population movement in some countries and regions [8,9,10], a remarkable decrease in the frequency of medical visits has been observed globally [2,3,4]. For example, there was an average reduction of 60.9% in planned hospital visits by June 2020 in the UK [3]. Due to the expected surge in hospitalizations and high demand for resources and capacity to effectively respond to the COVID-19 pandemic, there was an increase in care deferrals overall [11]; with all non-urgent elective surgeries being temporarily postponed in many countries around the world [12,13]. In Australia, these measures resulted in a substantial decrease in healthcare activity from March to June 2020, with a decrease of 51.5% in breast cancer screening activity and 32.6% in public hospital planned surgical activity [14]. Similar issues have been reported in other high-income countries [15,16,17,18]. China’s healthcare system was the first to be hit by the COVID-19 pandemic. The total medical visits decreased by 27.2% in the first quarter of 2020 in comparison to the same period of 2019 [5], with outpatient visits, inpatients visits, and emergency department visits all dropping sharply [19,20,21,22]. Most existing studies in China on the impact of COVID-19 on individuals’ healthcare-seeking behavior have been conducted at tertiary hospitals that are mainly located in urban areas. For example, by April 2020, pediatric outpatient visits in Beijing city slumped by 83% [21] and inpatients’ visits due to sorts of dermatopathies declined by 60.79% in Wuhan city [19]. Although rural residents in China account for 36% of the population [23] and have higher utilization of health services than urban residents [24], few studies have illustrated the impact of the COVID-19 pandemic on rural residents’ healthcare-seeking behavior [25]. The neglect of healthcare-seeking behavior and utilization of health services in rural areas during the COVID-19 pandemic has also been reported in other countries [26,27,28].

In China, rural residents have a slightly higher demand for health services than urban residents. For example, in 2018, the two-week outpatient visiting rates for rural and urban residents were 24.8% and 23.2%, and the annual hospitalization rates were 14.7% and 12.9%, respectively [24]. However, the quality of medical services in rural China is very limited due to huge inequalities in medical resources between rural and urban areas [29]. China’s healthcare system consists of three levels of medical institutions: primary healthcare facilities, secondary hospitals, and tertiary hospitals. Primary healthcare facilities, with 20–100 beds, are tasked with providing rehabilitation and nursing services for clearly diagnosed patients with chronic diseases, as well as basic healthcare for some common and frequently occurring diseases. Secondary hospitals, with 100–500 beds, provide comprehensive healthcare for patients with some perplexing diseases. Tertiary hospitals, with over 500 beds, provide specialist healthcare for patients with critical and serious diseases [30]. Primary healthcare facilities and secondary hospitals account for a large proportion of rural China [29]. Therefore, most medical institutions in rural areas are incapable of diagnosing and treating serious diseases, and rural residents are increasingly seeking medical care in large urban hospitals. It has been observed that the out-of-county hospitalization rate climbed from 12.4% in 2008 to 19.2% in 2016 [31]. However, the COVID-19 pandemic had a great impact on the healthcare system and long-distance traveling. For instance, there was a 49.9% decrease in average traveling distance in the first quarter of 2020 compared to the same period in 2019 [32], but such a decline could be a result of many factors. It could be that the location where rural residents seek medical services changes or travel to seek specialized care at different locations over time, but these issues are poorly documented in the literature. Understanding the interaction between these factors is important for promoting health equity between rural and urban areas.

To fill this gap, we aimed to analyze changes in rural residents’ hospitalization visits in different areas on the basis of the hospitalization location for different diseases via a controlled interrupted time series (CITS) approach. The results shed light on how the healthcare utilization of rural patients with different diseases distributed during the COVID-19 period and call for initiatives relating to policy, legislation, and regulations to tackle diseases, especially non-communicable diseases, in the future.

## 2. Materials and Methods

### 2.1. Data Sources

De-identified hospitalization data covering the period from April 2017 to March 2021 were extracted from the health information system in Yangxi County in Yangjiang City, Guangdong Province, which has 550,000 residents. Yangxi County is a county-level administrative region (the third-level administrative division in China), and Yangjiang City is a prefecture-level administrative region (the second-level administrative division) [33]. The health information system was constructed by the health insurance bureau of Yangxi County and collects medical records, including inpatient and outpatient information, of all national health insurance enrollees in the county. The information we collected from the health information system included patients’ age, gender, disease diagnoses, date of admission, date of discharge, hospitalization locations, and medical expenses. By 2018, national health insurance schemes had covered 97.6% of residents in rural areas in China [24].

Detailed information on Guangdong Province can be found at http://stats.gd.gov.cn/ (accessed on 1 September 2022).

### 2.2. CITS Design

Interrupted time series (ITS) design is the strongest quasi-experimental approach to assess the impact of interventions when a randomized controlled trial is not feasible [34]. However, ITS studies cannot exclude additional time-varying confounders that may affect the outcome [35]. The use of control in ITS studies, i.e., CITS, can limit the threat [36]. Segmented regression analysis is a powerful statistical method for estimating intervention effects in interrupted time series studies [37]. By modeling the underlying trend, the short- and long-term effects of the intervention are evaluated by changes in the level and slope of the time series, respectively [37].

In this study, a segmented regression analysis of the CITS approach [36] was used to estimate the effect of the COVID-19 pandemic on hospital visits among rural residents, using a historical control group [36]. The COVID-19 lockdown policy was implemented on 23 January 2020, in China. Therefore, in the CITS design, we chose February 2020 as the intervention time point. Then, we defined April 2019 to March 2021 as the intervention period, which has been further divided into two stages, namely, the pre-COVID period (April 2019 to January 2020) and the COVID-impacted period (February 2020 to March 2021). With the use of historical control under the ITS strategy, the control period was taken from April 2017 to March 2019, which was divided into two stages as well. April 2017 to January 2018 was defined as the equivalent pre-COVID period and February 2018 to March 2019 was defined as the equivalent COVID-impacted period.

### 2.3. Measurement

The dependent variable was the monthly number of hospital visits of residents in the study county. We measured the overall monthly number of hospital visits and hospital visits of different subgroups in three areas on the basis of the hospitalization location, i.e., within-county, out-of-county but within-city, and out-of-city. The definition of the spatial reference levels was based on divisions of administrative areas of Yangxi County and Yangjiang City. Subgroups included patients’ age (0–14/15–44/45–59/60 years and older), gender, level of the hospital (primary healthcare facilities/secondary hospitals/tertiary hospitals), and disease classification. Disease classification was defined according to the WHO International Classification of Diseases (ICD-10) codes [38] on the basis of inpatients’ disease diagnoses.

### 2.4. Statistical Analysis

We performed the CITS analyses on changes in rural residents’ hospitalization visits in three areas, using data from April 2017 to March 2021. In our study, we constructed the model as follows:Y_t_ = β_0_ + β_1_ Time + β_2_ Phase + β_3_ Post + β_4_ Group + β_5_ Group × Time + β_6_ Group × Phase + β_7_ Group × Post + Month + e_t_
(1)
where “Y_t_” is the monthly number of hospital visits; “Time” is the time point of data within a phase (0, 1, 2, 3, until 23); “Phase” indicates “pre-COVID/COVID-impacted period or the equivalent period” (0 refers to the pre-COVID period and the equivalent pre-COVID period and 1 refers to the COVID-impacted period and the equivalent COVID-impacted period); “Post” indicates the time point after the lockdown, the value is 0 before the lockdown and when the lockdown occurs, and then the value is taken from 1, 2, until 13; “Group” is the indicator variable for grouping (0 for the control period and 1 for the intervention period); “Group × Time” is an interaction term to account for the slope difference between the pre-COVID period and equivalent pre-COVID period; “Group × Phase” is an interaction term to account for the difference in the change of levels between the implementation period and control period; “Group × Post” is an interaction term to account for the difference in the change of slopes between the intervention period and control period; and “e_t_” denotes the random error.

For illustration, Figure 1 visualizes the interpretation of the coefficients in the equation 1, and two coefficients are important for evaluating the impact of COVID-19. As is seen in Figure 1, “β_6_” indicates the difference of change in the average level of monthly hospital visits between intervention and control periods (instantaneous effect of the COVID-19) and “β_7_” indicates the difference of the change in slope between intervention and control periods (the long-term effect of the COVID-19). Other coefficients are described in the Appendix A (Appendix A). Appendix A in Appendix A illustrates the data structure for the analyses of the effect of COVID-19. Subgroup CITS analyses were also conducted, in terms of gender, age, level of the hospital, and disease classification of inpatients.

We used negative binomial regressions to test the trend in monthly hospital visits after the COVID-19 lockdown policy because we confirmed the over-dispersion of data using the likelihood ratio tests, which violated the assumption of Poisson regression [39]. To adjust for autocorrelation and heteroscedasticity of residuals, we used the Newey–West method [40] to estimate standard errors. We also included monthly indicator variables as covariates to adjust for observed seasonal patterns [37]. All statistical analyses were performed using R (version 4.0.2). All tests were two-tailed, and statistical significance was defined as *p* < 0.05.

## 3. Results

A total of 262,847 hospital visits (62,706 in the pre-COVID period and 68,009 in the COVID-impacted period; 50,910 in the equivalent pre-COVID period and 81,222 in the equivalent COVID-impacted period) were included in this study. Table 1, Table 2 and Table 3 show the number of monthly hospital visits before and after the COVID-19 lockdown policy in February 2020 and during the equivalent control periods in within-county, out-of-county but within-city, and out-of-city areas, respectively.

### 3.1. Changes in the Monthly Number of Hospital Visits in Three Hospitalization Areas

After the lockdown policy, as shown in Figure 2, Figure 3 and Figure 4 and Table 4, there were immediate decreases (β_6_) in monthly hospital visits in the three areas (within-city: −43.77%, 95% CI: −57.78%, −29.72%; out-of-county but within-city: −38.28%, (−46.38%, −30.17%); out-of-city: −24.28%, (−32.17%, −16.38%)). However, compared to the control period, an increasing trend (β_7_) in monthly hospital visits was only found in the out-of-county but within-city area (2.81%, (1.96%, 3.66%)).

### 3.2. Instantaneous Effect of COVID-19 According to Subgroup Analyses

Subgroup analyses in three different hospitalization areas are shown in Figure 5, Figure 6 and Figure 7. Immediate decreases in the number of hospital visits (β_6_) were observed in most subgroups in the three hospitalization areas, especially for primary healthcare facilities (−212.58%, (−393.44%, −31.71%)) and secondary hospitals (−209.88%, (−280.44%, −139.33%)) in the out-of-county but within-city area.

In the within-county area, the diseases of the respiratory system (J00–J99); endocrine, nutritional, and metabolic diseases (E00–E90); and mental and behavioral disorders (F00–F99), ranking as the top three decreased cause-specific hospital visits, remarkably decreased by 114.14%, 102.53%, and 90.65% compared with the hospital visits during the control period, respectively. In the out-of-county but within-city area, the diseases of the ear and mastoid process (H60–H95), certain infectious and parasitic diseases (A00–B99), and the disease of the respiratory system (J00–J99), ranking as the top three decreased cause-specific hospital visits, remarkably decreased by 153.57%, 95.18%, and 90.37% compared with the hospital visits during the control period, respectively. In the out-of-city area, the diseases of the ear and mastoid process (H60–H95), certain infectious and parasitic diseases (A00–B99), and certain conditions originating in the perinatal period (P00–P96), ranking as the top three decreased cause-specific hospital visits, remarkably decreased by 172.61%, 133.57%, and 81.01% compared with the hospital visits during the control period, respectively.

### 3.3. Long-Term Impacts of COVID-19 According to Subgroup Analyses

As shown in Figure 5, Figure 6 and Figure 7, there was an acceleration (β_7_) in monthly hospital visits after COVID-19 lockdown for both genders (males 4.58%, (3.60%, 5.56%); females 1.20%, (0.09%, 2.31%)) and people aged 45–59 years (8.28%, (3.98%, 12.59%)) in the out-of-county but within-city area.

Moreover, in the subgroup analysis based on disease classification, we identified three major patterns of change in healthcare-seeking behaviors over time. The first pattern was the acceleration in monthly hospital visits in the out-of-county but within-city area and deceleration in within-county and/or out-of-city areas after the lockdown. This kind of pattern was reflected in some common non-communicable diseases (NCDs). The monthly number of hospital visits due to neoplasms (C00–D48), diseases of the circulatory system (I00–I99), and diseases of the musculoskeletal system and connective tissue (M00–M99) were accelerating in the out-of-county but within-city area (C00–D48: 2.03%, (0.25%, 3.82%); I00–I99: 3.26%, (2.09%, 4.43%); M00–M99: 11.01%, (6.58%, 15.45%)) but decelerating in the within-county area (C00–D48: −4.83%, (−6.20%, −3.45%); I00–I99: −1.03%, (−1.99%, −0.08%); M00–M99: −2.32%, (−3.48%, −1.16%)). Moreover, the deceleration remained significant for I00-I99 in the out-of-city area as well (−2.12%, (−3.67%, −0.57%)). The monthly number of hospital visits for the endocrine, nutritional, and metabolic diseases (E00–E90) and injury, poisoning, and certain other consequences of external causes (S00–T98) were accelerating in the out-of-county but within-city area (E00–E90: 3.89%, (1.46%, 6.32%); S00-T98: 5.06%, (2.34%, 7.78%)) but decelerating in the out-of-city area (E00–E90: −8.19%, (−12.50%, −3.87%); S00–T98: −9.95%, (−12.22%, −7.68%)).

The second pattern was that monthly hospital visits decelerated in all three areas after the lockdown. This pattern can be seen in diseases of pregnancy, childbirth, and the puerperium (O00–O99); mental and behavioral disorders (F00–F99); and diseases of the nervous system (G00-G99). The monthly number of hospital visits for the O00–O99 were decelerating in all the within-county area (−12.16%, (−13.84%, −10.47%)), out-of-county but within-city area (−4.78%, (−6.14%, −3.41%)), and out-of-city area (−10.01%, (−12.47%, −7.54%)). The monthly numbers of hospital visits for the F00–F99 (−4.05%, (−6.19%, −1.91%)) and the G00–G99 (−3.88%, (−7.02%, −0.74%)) were significantly decelerating in the out-of-county but within-city area while showed a nonsignificant decelerating trend in other areas.

The third pattern was that monthly hospital visits accelerated in all three areas after the lockdown. This kind of pattern can be seen in the diseases of the eye and adnexa (H00–H59). The monthly number of hospital visits for the H00–H59 was accelerating in all the within-county area (16.04%, (11.78%, 20.30%)), out-of-county but within-city area (9.31%, (3.72%, 14.91%)), and out-of-city area (7.35%, (2.06%, 12.65%)).

## 4. Discussion

The pattern of health-seeking behaviors differed markedly by disease classification, and we identified three major patterns. The most predominant pattern of change in healthcare-seeking behaviors after the lockdown was an increase in hospitalizations in the out-of-county but within-city area, but a decrease in hospitalizations in other areas. This pattern was observed in common NCDs, such as neoplasms; diseases of the circulatory system; and endocrine, nutritional, and metabolic diseases that accounted for 28.2% of the total hospitalizations. This pattern could be explained by two main reasons. Firstly, the prevalence of chronic diseases in rural areas in China is accelerating and higher than that in urban areas [24]. However, unlike high-level hospitals, the primary healthcare facilities, which account for a great part of healthcare facilities in rural areas, do not have sufficient drugs, medical supplies, and equipment to treat chronic diseases and conditions including cancers [41]. Therefore, to seek comprehensive and high-quality medical services, rural patients with chronic diseases tended to go outside the county where there are more quality medical services [42,43]. Secondly, because public transportation is considered a potential hotspot for COVID-19 transmission [44], coupled with travel restrictions [45], rural residents may prefer not to travel too far for medical care. As a result, rural residents’ demand for hospitalization for NCDs with a high disease burden was redistributed across areas compared to the control period. This phenomenon suggested that although China has made tremendous efforts to improve the quality of health care in rural areas [46], improving the capacity and quality of medical services in rural areas remains an important goal of China’s future health care reform.

Secondly, we found that for diseases of pregnancy, childbirth, and the puerperium, as well as for mental, behavioral, and neurological disorders, inpatient visits showed a downward trend in all areas. The decrease in pregnancy, childbirth, and puerperium hospitalizations was largely attributable to the significant decline in fertility intentions and fertility rates in China in recent years. Due to changes in views regarding fertility and gender equality, as well as the increase in the cost of raising children [47], China’s total fertility rate of women of childbearing age has continued to decline in recent years. The total fertility rate was 1.3 according to the results of the latest national population census [48]. China only recorded 10.6 million births in 2021 [49], marking a drop for the fifth consecutive year. Moreover, the impact of COVID-19 on male and female fertility has also attracted academic attention, but research findings are still unclear and further research is needed [50,51].

More importantly, although previous studies indicated that the prevalence of mental health problems during COVID-19 has dramatically increased worldwide in the long term [52,53], we found that hospital visits for diseases such as mental, behavioral, and neurological disorders declined among rural residents in China. The study findings suggested that unlike the redistribution of demand for inpatient services across areas for other NCDs, the overall demand for medical care for mental illness among rural residents in China has been suppressed during the pandemic. Similarly, in most settings in low- and middle-income countries (LMICs), inpatient psychiatric facilities were temporarily repurposed to treat patients with COVID-19, and patients with severe mental illness are often unable to obtain medications or attend treatment facilities [53,54]. In South Africa, seeking psychiatric care from hospital facilities has decreased because of concerns about SARS-CoV-2 infection and stigma [55]. This suggested that both individuals and policymakers should take the initiatives to address the mental health crisis during the COVID-19 pandemic [56]. From the perspective of individuals, individuals can take actions, such as exercising regularly and maintaining a healthy diet pattern, as well as keeping in touch with friends and family by phone calls, to effectively ease and prevent symptoms of depression [57,58,59,60]. From the perspective of policymakers, with very limited mental health professionals in LMICs [61], we urgently require strong intervention policies, such as building capacities among non-specialists to deliver psychological services, strengthening remote mental health services, and implementing digital mental health interventions [62,63] in order to address the need for mental health services in resource-poor areas to avoid exacerbating health inequities and to move towards achieving SDG 3: Good Health and Well-Being [64].

Thirdly, we found that hospital visits of diseases of the eye and adnexa restored rapidly in all three areas after the lockdown. Dental, ophthalmology, and otolaryngology departments are high-risk departments for COVID-19 infection [65,66,67]. In the early stage of the COVID-19 pandemic, such departments in China were suspended or temporarily repurposed to treat patients with COVID-19 [68]. The restoration of such kinds of diseases can be explained by the restoration of cataract surgery during the COVID-19 pandemic. Cataract surgery is the commonest ophthalmic surgery, and it can be performed at various levels of the hospital [69]. Patients’ demand for cataract surgery did not decrease during the pandemic [70], and therefore, most cataract surgery patients who were a vulnerable age were willing to attend elective cataract surgery due to worsening eyesight when the pandemic was contained [71]. The restoration outside the city can be further explained by the location of the study county. Yangxi County is at the junction of two cities in Guangdong Province. Considering the short length of hospital stay for cataract surgery, normally 2–3 days, it is convenient for cataract patients to receive cataract surgery in hospitals across the city.

This study has several limitations. First, because the data were extracted from the Health Information System that only has disease diagnoses but lacks information on disease severity, we could not further examine changes in healthcare utilization of rural residents according to disease severity. Second, hospital visits in our study showed a downward trend at the end of each year, and this mainly resulted from the concurrent health insurance payment policy. Under the current health insurance payment policy, hospitals are provided a fixed amount of annual health insurance funding annually. To control the total health care cost and retain the surplus of funding, hospitals might enhance the hospitalization criteria at the end of the year. In our study, we took historical cohort control to rule out the effect of this policy that occurs on an annual basis.

## 5. Conclusions

This study is the first to explore the impact of COVID-19 on the healthcare-seeking behavior patterns of rural patients on the basis of data from a rural county, where healthcare resources are scarce, in Guangdong Province, China. This study revealed that the pattern of hospitalization utilization differed markedly by disease classifications, and we identified three major patterns. Our findings hint that targeted strategies and policies are urgently needed in resource-poor areas, such as rural China, to ensure that people have access to adequate, high-quality healthcare services, particularly mental and neurological healthcare, during the COVID-19 pandemic. Moreover, rural patients with mental and neurological illnesses should also take the initiatives to ease and prevent stress. Future studies may need to further identify the patterns of health-seeking behaviors of rural patients with mental and neurological illnesses whose demand for medical care has been suppressed with the consideration of disease severity, as well as to explore the reasons why their health-seeking behaviors have changed.

## Figures and Tables

**Figure 1 ijerph-19-13259-f001:**
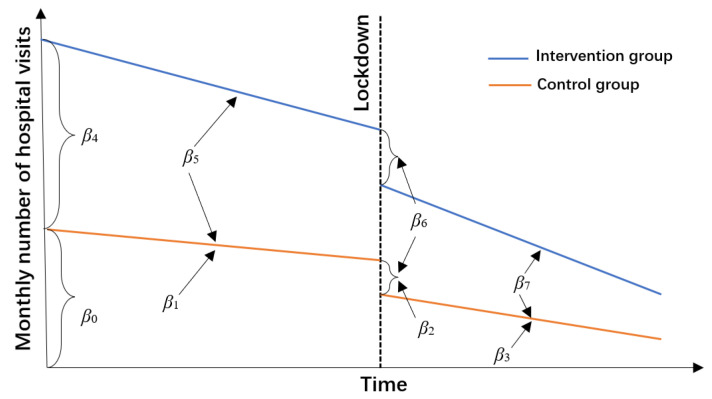
The segmented regression model for a controlled interrupted time series analysis.

**Figure 2 ijerph-19-13259-f002:**
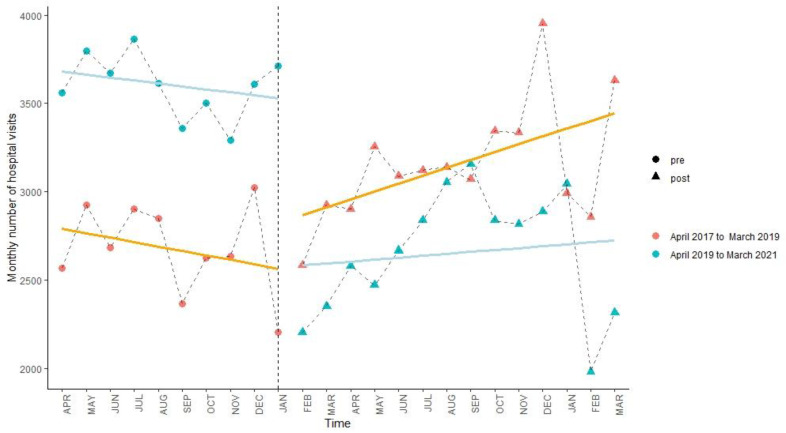
Interrupted time series analysis of hospital visits within Yangxi County from April 2017 to March 2019 (orange line) and from April 2019 to March 2021 (blue line). The line is the model fit and the vertical dashed line indicates the intervention (23 January 2020).

**Figure 3 ijerph-19-13259-f003:**
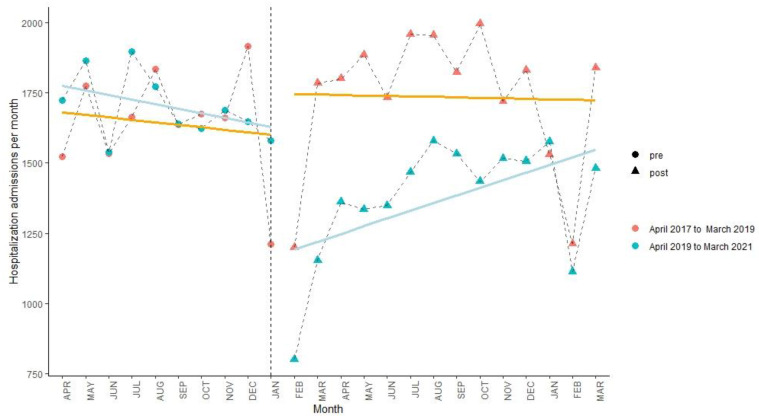
Interrupted time series analysis of hospital visits outside Yangxi County but still within Yangjiang City from April 2017 to March 2019 (orange line) and from April 2019 to March 2021 (blue line). The line is the model fit and the vertical dashed line indicates the intervention (23 January 2020).

**Figure 4 ijerph-19-13259-f004:**
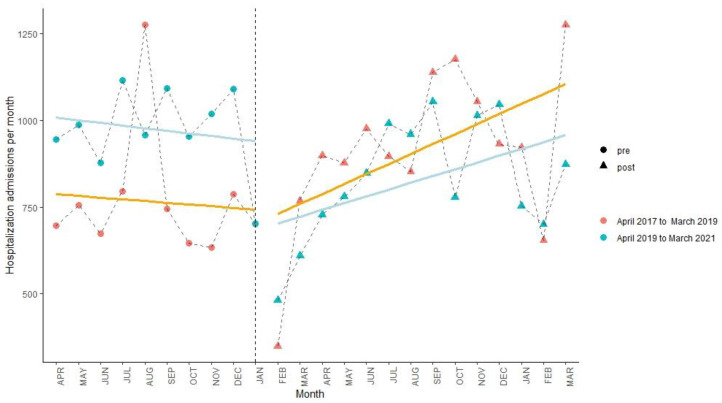
Interrupted time series analysis of hospital visits outside Yangjiang City from April 2017 to March 2019 (orange line) and from April 2019 to March 2021 (blue line). The line is the model fit and the vertical dashed line indicates the intervention (23 January 2020).

**Figure 5 ijerph-19-13259-f005:**
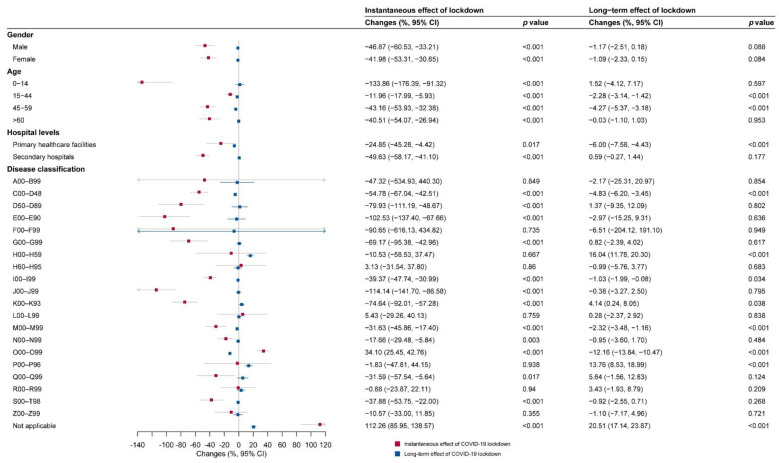
The instantaneous and long-term effect of COVID-19 lockdown on hospital visits within the county for subgroups. A00–B99: certain infectious and parasitic diseases; C00–D48: neoplasms; D50–D89: diseases of the blood and blood-forming organs and certain disorders involving the immune mechanism; E00–E90: endocrine, nutritional, and metabolic diseases; F00–F99: mental and behavioral disorders; G00–G99: diseases of the nervous system; H00–H59: diseases of the eye and adnexa; H60–H95: diseases of the ear and mastoid process; I00–I99: diseases of the circulatory system; J00–J99: diseases of the respiratory system; K00–K93: diseases of the digestive system; L00–L99: diseases of the skin and subcutaneous tissue; M00–M99: diseases of the musculoskeletal system and connective tissue; N00–N99: diseases of the genitourinary system; O00–O99: pregnancy, childbirth, and the puerperium; P00–P96: certain conditions originating in the perinatal period; Q00–Q99: congenital malformation deformations and chromosomal abnormality; R00–R99: symptoms, signs, and abnormal clinical and laboratory findings, not elsewhere classified; S00–T98: injury, poisoning, and certain other consequences of external causes; Z00–Z99: factors influencing health status and contact with health services; not applicable: not classified.

**Figure 6 ijerph-19-13259-f006:**
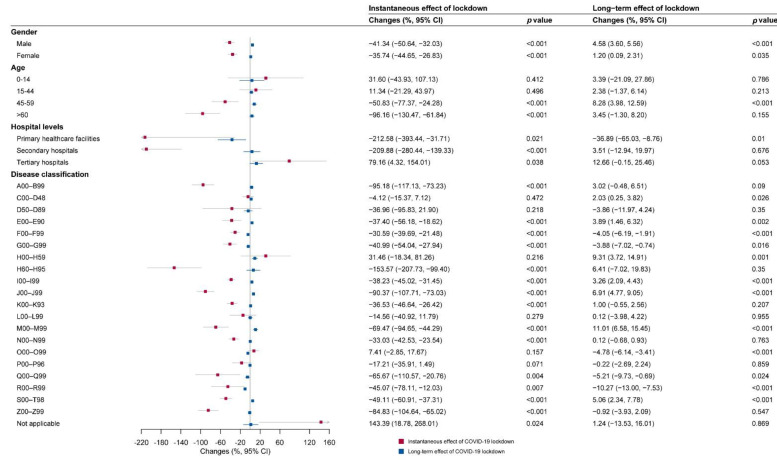
The instantaneous and long-term effect of COVID-19 lockdown on hospital visits within the city but outside the county for subgroups. A00–B99: certain infectious and parasitic diseases; C00–D48: neoplasms; D50–D89: diseases of the blood and blood-forming organs and certain disorders involving the immune mechanism; E00–E90: endocrine, nutritional, and metabolic diseases; F00–F99: mental and behavioral disorders; G00–G99: diseases of the nervous system; H00–H59: diseases of the eye and adnexa; H60–H95: diseases of the ear and mastoid process; I00–I99: diseases of the circulatory system; J00–J99: diseases of the respiratory system; K00–K93: diseases of the digestive system; L00–L99: diseases of the skin and subcutaneous tissue; M00–M99: diseases of the musculoskeletal system and connective tissue; N00–N99: diseases of the genitourinary system; O00–O99: pregnancy, childbirth, and the puerperium; P00–P96: certain conditions originating in the perinatal period; Q00–Q99: congenital malformation deformations and chromosomal abnormality; R00–R99: symptoms, signs, and abnormal clinical and laboratory findings, not elsewhere classified; S00–T98: injury, poisoning, and certain other consequences of external causes; Z00–Z99: factors influencing health status and contact with health services; not applicable: not classified.

**Figure 7 ijerph-19-13259-f007:**
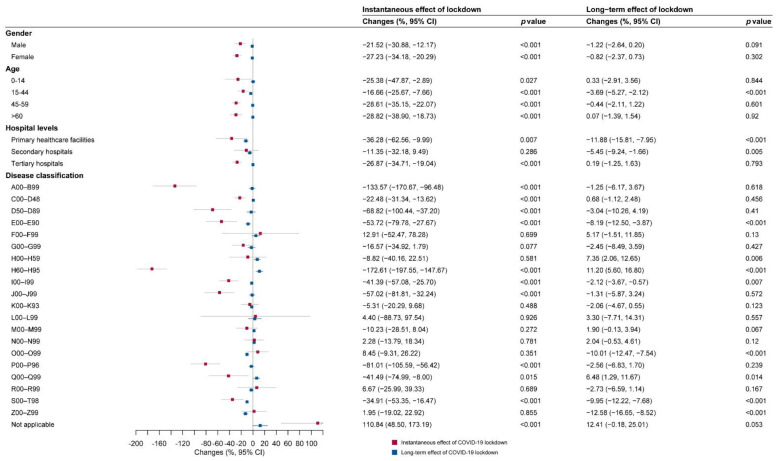
The instantaneous and long-term effect of COVID-19 lockdown on hospital visits outside the city for subgroups. A00–B99: certain infectious and parasitic diseases; C00–D48: neoplasms; D50–D89: diseases of the blood and blood-forming organs and certain disorders involving the immune mechanism; E00–E90: endocrine, nutritional, and metabolic diseases; F00–F99: mental and behavioral disorders; G00–G99: diseases of the nervous system; H00–H59: diseases of the eye and adnexa; H60–H95: diseases of the ear and mastoid process; I00–I99: diseases of the circulatory system; J00–J99: diseases of the respiratory system; K00–K93: diseases of the digestive system; L00–L99: diseases of the skin and subcutaneous tissue; M00–M99: diseases of the musculoskeletal system and connective tissue; N00–N99: diseases of the genitourinary system; O00–O99: pregnancy, childbirth, and the puerperium; P00–P96: certain conditions originating in the perinatal period; Q00–Q99: congenital malformation deformations and chromosomal abnormality; R00–R99: symptoms, signs, and abnormal clinical and laboratory findings, not elsewhere classified; S00–T98: injury, poisoning, and certain other consequences of external causes; Z00–Z99: factors influencing health status and contact with health services; not applicable: not classified.

**Table 1 ijerph-19-13259-t001:** Average monthly hospital visits in within-county areas for rural residents, shown as mean (SD).

	Implementation Period			Control Period		
	Pre-COVID (a)	COVID-Impacted (b)	Change 1 (%): (b − a)/a	Equivalent Pre-COVID (c)	Equivalent COVID-Impacted (d)	Change 2 (%): (d − c)/c
Overall	3599 (171)	2657 (339)	−26.18	2677 (245)	3156 (328)	17.89
Gender						
Female	1767 (125)	1363 (201)	−22.90	1164 (131)	1474 (167)	26.64
Male	1832 (60)	1294 (147)	−29.35	1514 (134)	1682 (170)	11.16
Age (years)						
0–14	471 (67)	222 (69)	−52.82	368 (81)	421 (84)	14.62
15–44	816 (57)	629 (67)	−22.92	805 (84)	812 (83)	0.84
45–59	565 (30)	402 (55)	−28.75	345 (46)	449 (66)	30.40
≥60	1747 (109)	1403 (212)	−19.70	1160 (142)	1474 (184)	27.04
Hospital levels						
Primary healthcare facilities	1035 (90)	778 (224)	−24.88	844 (115)	1030 (137)	22.00
Secondary hospitals	2564 (136)	1879 (269)	−26.71	1833 (159)	2127 (223)	16.00
Tertiary hospitals	0 (0)	0 (0)	-	0 (0)	0 (0)	-
Disease classification (ICD–10 code)						
A00–B99	114 (30)	67 (24)	−41.14	81 (24)	92 (26)	14.16
C00–D48	169 (17)	147 (23)	−13.01	94 (15)	120 (19)	27.78
D50–D89	53 (8)	45 (9)	−15.77	17 (4)	48 (9)	177.87
E00–E90	104 (16)	88 (25)	−15.36	30 (7)	77 (19)	160.22
F00–F99	11 (2)	8 (3)	−33.63	4 (2)	7 (3)	73.21
G00–G99	151 (21)	115 (24)	−23.95	37 (9)	98 (21)	165.86
H00–H59	75 (10)	62 (21)	−17.08	76 (27)	70 (27)	−7.28
H60–H95	15 (5)	9 (3)	−40.72	16 (6)	16 (4)	0.53
I00–I99	776 (53)	606 (76)	−21.96	514 (65)	683 (104)	32.81
J00–J99	698 (114)	341 (102)	−51.20	500 (114)	587 (115)	17.46
K00–K93	359 (44)	199 (25)	−44.65	230 (43)	352 (35)	53.12
L00–L99	43 (10)	39 (13)	−10.14	25 (9)	31 (8)	24.14
M00–M99	226 (27)	153 (32)	−32.09	173 (27)	192 (30)	11.22
N00–N99	209 (20)	165 (22)	−21.34	103 (16)	161 (39)	56.32
O00–O99	287 (32)	243 (76)	−15.39	465 (77)	322 (33)	−30.80
P00–P96	50 (10)	135 (73)	171.26	53 (13)	51 (9)	−4.14
Q00–Q99	11 (2)	11 (4)	1.08	4 (2)	8 (2)	120.08
R00–R99	46 (11)	37 (7)	−21.18	100 (28)	56 (12)	−43.97
S00–T98	154 (13)	137 (21)	−11.34	94 (18)	126 (17)	34.68
Z00–Z99	39 (7)	41 (7)	5.96	33 (7)	47 (15)	40.49
Not applicable	9 (4)	12 (6)	32.31	31 (11)	13 (7)	−56.21

Implementation period: period from April 2019 to March 2021; control period: period from April 2017 to March 2019; pre-COVID period: period from April 2019 to January 2020; COVID-impacted period: period form February 2020 to March 2021; equivalent pre-COVID period: period from April 2017 to January 2018; equivalent COVID-impacted period: period from February 2018 to March 2019; A00–B99: certain infectious and parasitic diseases; C00–D48: neoplasms; D50–D89: diseases of the blood and blood-forming organs and certain disorders involving the immune mechanism; E00–E90: endocrine, nutritional, and metabolic diseases; F00–F99: mental and behavioral disorders; G00–G99: diseases of the nervous system; H00–H59: diseases of the eye and adnexa; H60–H95: diseases of the ear and mastoid process; I00–I99: diseases of the circulatory system; J00–J99: diseases of the respiratory system; K00–K93: diseases of the digestive system; L00–L99: diseases of the skin and subcutaneous tissue; M00–M99: diseases of the musculoskeletal system and connective tissue; N00–N99: diseases of the genitourinary system; O00–O99: pregnancy, childbirth, and the puerperium; P00–P96: certain conditions originating in the perinatal period; Q00–Q99: congenital malformation deformations and chromosomal abnormality; R00–R99: symptoms, signs, and abnormal clinical and laboratory findings, not elsewhere classified; S00–T98: injury, poisoning, and certain other consequences of external causes; Z00–Z99: factors influencing health status and contact with health services; not applicable: not classified.

**Table 2 ijerph-19-13259-t002:** Average monthly hospital visits in out-of-county but within-city areas for rural residents, shown as mean (SD).

	Implementation Period			Control Period		
	Pre-COVID (a)	COVID-Impacted (b)	Change 1 (%): (b − a)/a	Equivalent Pre-COVID (c)	Equivalent COVID-Impacted (d)	Change 2 (%): (d − c)/c
Overall	1698 (112)	1372 (210)	−19.18	1643 (186)	1734 (243)	5.53
Gender						
Female	781 (59)	645 (115)	−17.39	775 (89)	797 (125)	2.90
Male	917 (60)	727 (98)	−20.70	868 (109)	936 (125)	7.87
Age (years)						
0–14	183 (63)	124 (77)	−32.20	187 (33)	211 (32)	13.04
15–44	458 (94)	433 (208)	−5.37	487 (60)	487 (57)	0.09
45–59	369 (46)	355 (139)	−3.87	328 (40)	355 (55)	8.44
≥60	688 (133)	460 (165)	−33.10	642 (91)	680 (130)	5.99
Hospital levels						
Primary healthcare facilities	111 (94)	30 (56)	−73.13	110 (19)	78 (36)	−28.77
Secondary hospitals	515 (191)	279 (413)	−45.87	207 (48)	472 (131)	127.84
Tertiary hospitals	1019 (379)	1047 (514)	−2.80	1204 (144)	1118 (124)	−7.17
Disease classification (ICD–10 codes)						
A00–B99	38 (8)	19 (5)	−48.76	42 (10)	41 (10)	−4.03
C00–D48	201 (17)	185 (32)	−7.89	170 (27)	161 (31)	−5.32
D50–D89	23 (5)	18 (3)	−20.57	31 (7)	23 (5)	−26.74
E00–E90	47 (7)	37 (9)	−21.58	40 (4)	44 (11)	9.81
F00–F99	74 (25)	127 (27)	72.65	42 (12)	55 (12)	29.18
G00–G99	68 (16)	64 (11)	−6.20	33 (11)	42 (10)	26.13
H00–H59	84 (31)	56 (21)	−33.85	96 (45)	66 (24)	−30.89
H60–H95	12 (6)	8 (3)	−33.88	13 (4)	18 (6)	36.21
I00–I99	200 (24)	158 (23)	−20.83	249 (35)	255 (44)	2.34
J00–J99	211 (38)	104 (28)	−50.85	196 (35)	214 (38)	9.53
K00–K93	176 (16)	137 (23)	−21.99	178 (25)	180 (24)	0.90
L00–L99	17 (3)	14 (4)	−14.54	13 (3)	15 (4)	19.60
M00–M99	131 (30)	89 (21)	−32.22	142 (25)	205 (91)	44.21
N00–N99	154 (17)	112 (26)	−27.22	142 (22)	152 (25)	7.07
O00–O99	81 (10)	74 (10)	−9.26	101 (13)	94 (10)	−6.28
P00–P96	41 (7)	33 (6)	−18.94	32 (7)	37 (8)	15.57
Q00–Q99	14 (6)	11 (7)	−21.58	11 (7)	11 (5)	−1.63
R00–R99	16 (3)	14 (5)	−14.82	20 (8)	15 (4)	−24.91
S00–T98	57 (7)	48 (11)	−15.07	48 (10)	55 (10)	14.02
Z00–Z99	51 (12)	59 (12)	16.16	37 (5)	46 (23)	25.54
Not applicable	3 (3)	5 (2)	65.90	6 (4)	4 (3)	−26.15

Implementation period: period from April 2019 to March 2021; control period: period from April 2017 to March 2019; pre-COVID period: period from April 2019 to January 2020; COVID-impacted period: period form February 2020 to March 2021; equivalent pre-COVID period: period from April 2017 to January 2018; equivalent COVID-impacted period: period from February 2018 to March 2019; A00–B99: certain infectious and parasitic diseases; C00–D48: neoplasms; D50–D89: diseases of the blood and blood-forming organs and certain disorders involving the immune mechanism; E00–E90: endocrine, nutritional, and metabolic diseases; F00–F99: mental and behavioral disorders; G00–G99: diseases of the nervous system; H00–H59: diseases of the eye and adnexa; H60–H95: diseases of the ear and mastoid process; I00–I99: diseases of the circulatory system; J00–J99: diseases of the respiratory system; K00–K93: diseases of the digestive system; L00–L99: diseases of the skin and subcutaneous tissue; M00–M99: diseases of the musculoskeletal system and connective tissue; N00–N99: diseases of the genitourinary system; O00–O99: pregnancy, childbirth, and the puerperium; P00–P96: certain conditions originating in the perinatal period; Q00–Q99: congenital malformation deformations and chromosomal abnormality; R00–R99: symptoms, signs, and abnormal clinical and laboratory findings, not elsewhere classified; S00–T98: injury, poisoning, and certain other consequences of external causes; Z00–Z99: factors influencing health status and contact with health services; not applicable: not classified.

**Table 3 ijerph-19-13259-t003:** Average monthly hospital visits in the out-of-city area for rural residents, shown as mean (SD).

	Implementation Period			Control Period		
	Pre-COVID (a)	COVID-Impacted (b)	Change 1 (%): (b − a)/a	Equivalent Pre-COVID (c)	Equivalent COVID-Impacted (d)	Change 2 (%): (d − c)/c
Overall	974 (116)	829 (166)	−14.89	771 (176)	912 (221)	18.26
Gender						
Female	476 (46)	417 (74)	−12.33	371 (88)	434 (104)	17.04
Male	498 (72)	412 (94)	−17.33	400 (90)	478 (119)	19.39
Age (years)						
0–14	102 (14)	82 (16)	−18.87	72 (27)	88 (31)	22.98
15–44	252 (34)	200 (40)	−20.42	243 (52)	259 (61)	6.63
45–59	327 (42)	286 (59)	−12.60	248 (64)	307 (76)	23.48
≥60	294 (33)	260 (57)	−11.33	209 (40)	258 (62)	23.95
Hospital levels						
Primary healthcare facilities	17 (3)	14 (4)	−16.04	10 (4)	13 (5)	35.64
Secondary hospitals	175 (28)	155 (35)	−11.26	130 (47)	159 (57)	22.72
Tertiary hospitals	775 (91)	655 (134)	−15.43	625 (131)	733 (166)	17.25
Disease classification (ICD-10 codes)						
A00–B99	18 (5)	10 (4)	−44.84	19 (10)	19 (6)	1.66
C00–D48	271 (21)	247 (45)	−8.83	214 (24)	261 (47)	21.98
D50–D89	13 (3)	8 (3)	−35.43	7 (5)	9 (4)	25.00
E00–E90	22 (4)	17 (5)	−23.42	16 (6)	20 (9)	28.30
F00–F99	12 (5)	9 (4)	−22.41	13 (6)	14 (5)	6.32
G00–G99	23 (6)	20 (5)	−13.04	22 (10)	21 (8)	−2.90
H00–H59	19 (6)	16 (6)	−19.73	14 (6)	16 (6)	9.62
H60–H95	9 (3)	6 (2)	−34.32	4 (2)	7 (4)	64.97
I00–I99	134 (26)	123 (33)	−7.87	97 (21)	121 (37)	25.39
J00–J99	80 (13)	47 (11)	−41.22	51 (19)	72 (23)	41.31
K00–K93	75 (12)	65 (17)	−13.09	64 (17)	77 (25)	19.90
L00–L99	8 (2)	6 (3)	−25.75	6 (3)	5 (4)	−7.27
M00–M99	48 (9)	35 (8)	−27.52	41 (8)	41 (12)	0.52
N00–N99	79 (19)	54 (19)	−31.37	68 (21)	79 (21)	16.90
O00–O99	33 (6)	28 (5)	−14.51	41 (11)	40 (13)	−0.91
P00–P96	11 (4)	11 (4)	−2.36	10 (5)	12 (4)	22.24
Q00–Q99	15 (4)	13 (6)	−9.36	14 (5)	12 (5)	−14.81
R00–R99	21 (4)	17 (5)	−17.13	19 (6)	17 (6)	−10.43
S00–T98	23 (6)	19 (5)	−17.58	22 (6)	25 (10)	14.88
Z00–Z99	61 (8)	75 (16)	24.26	29 (12)	37 (16)	27.45
Not applicable	2 (2)	3 (2)	104.76	2 (2)	6 (4)	214.29

Implementation period: period from April 2019 to March 2021; control period: period from April 2017 to March 2019; pre-COVID period: period from April 2019 to January 2020; COVID-impacted period: period form February 2020 to March 2021; equivalent pre-COVID period: period from April 2017 to January 2018; equivalent COVID-impacted period: period from February 2018 to March 2019; A00–B99: certain infectious and parasitic diseases; C00–D48: neoplasms; D50–D89: diseases of the blood and blood-forming organs and certain disorders involving the immune mechanism; E00–E90: endocrine, nutritional, and metabolic diseases; F00–F99: mental and behavioral disorders; G00–G99: diseases of the nervous system; H00–H59: diseases of the eye and adnexa; H60–H95: diseases of the ear and mastoid process; I00–I99: diseases of the circulatory system; J00–J99: diseases of the respiratory system; K00–K93: diseases of the digestive system; L00–L99: diseases of the skin and subcutaneous tissue; M00–M99: diseases of the musculoskeletal system and connective tissue; N00–N99: diseases of the genitourinary system; O00–O99: pregnancy, childbirth, and the puerperium; P00–P96: certain conditions originating in the perinatal period; Q00–Q99: congenital malformation deformations and chromosomal abnormality; R00–R99: symptoms, signs, and abnormal clinical and laboratory findings, not elsewhere classified; S00–T98: injury, poisoning, and certain other consequences of external causes; Z00–Z99: factors influencing health status and contact with health services; not applicable: not classified.

**Table 4 ijerph-19-13259-t004:** Overall controlled interrupted time-series analysis: negative binomial regression on monthly hospital visits adjusted for seasonal effects.

Variable	Coefficient	Within-County Area		Out-Of-County but Within-City Area		Out-of-City Area	
		Changes (%, 95% CI)	*p*	Changes (%, 95% CI)	*p*	Changes (%, 95% CI)	*p*
Time	β_1_	−2.10 (−3.61, −0.57)	0.007	0.05 (−1.17, 1.27)	0.939	0.07 (−2.52, 2.65)	0.961
Phase	β_2_	24.22 (12.09, 36.34)	<0.001	10.03 (−0.76, 20.83)	0.069	−1.02 (−19.66, 17.62)	0.914
Post	β_3_	3.17 (1.63, 4.66)	<0.001	0.00 (−1.04, 1.04)	0.997	3.73 (1.26, 6.21)	0.003
Group	β_4_	27.79 (24.92, 30.64)	<0.001	5.77 (3.03, 8.52)	<0.001	24.52 (19.37, 29.67)	<0.001
Group × Time	β_5_	0.47 (−0.21, 1.06)	0.192	−0.55 (−1.27, 0.16)	0.129	−0.10 (−0.97, 0.77)	0.817
Group × Phase	β_6_	−43.77 (−57.78, −29.72)	<0.001	−38.28 (−46.38, −30.17)	<0.001	−24.28 (−32.17, −16.38)	<0.001
Group × Post	β_7_	−1.11 (−2.47, 0.27)	0.114	2.81 (1.96, 3.66)	<0.001	−1.04 (−2.52, 0.44)	0.167

CI: confidence interval.

## Data Availability

The data are available upon request from the authors.

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
