# Peer review of "Long-Term Impact of COVID-19 on Hospital Visits of Rural Residents in Guangdong, China: A Controlled Interrupted Time Series Study"

_ijerph, 2022, doi:10.3390/ijerph192013259_

Round 1
Reviewer 1 Report
Dear Authors,
Thank you very much for these enlightening insights into the impact of COVID-19 on hospital visits of rural inhabitants of Guangdong province. In the following, please find my comments and recommendations.
Title: In my opinion, you address to hospital visits of rural residents
Abstract: The abstract is too long. Please insert a paragraph on the discussion of your findings.
Keywords: Avoid to repeat title words.
Introduction: What do you mean by “internet hospitals”? Please, rephrase.
Line 43: “in the post-pandemic era”: Shouldn´t it say “during the pandemic-era”? Please, explain.
Line 45: Insert citation at the end of the sentence.
Line 79: With this sentence you describe the impact on long-distance travelers. In the next sentence you describe the average migration distance. Shouldn´t it say “commuting distance”? [[migration = change of place of residence]
Please, insert a paragraph on inpatient healthcare in China. This is necessary in order to understand the need for “targeted strategies and policies … in resource-poor areas” (line 439f).
At the end of this section please insert a paragraph on the aim and structure of the paper.
Material and Methods:
Please, shift last paragraph from introduction section to the beginning of this chapter.
Please, describe the study design more accurate.
Please, provide a separate descriptive paragraph on the case-study region Guangdong province (incl. surface, number of residents of the counties; number and names of cities and rural municipalities/counties). A map would be fine.
Please, provide more information on the health information system (line 98).
CITS approach: please, describe why you decided for this methodology. Furthermore, please explain the determination of the time-periods (pre-COVID, COVID-impacted period).
Line 114: What do you mean by “main outcome”? Shouldn´t it say “dependent variable”?
Please provide a sub-chapter on the “segmented regression interaction model”. This term need explanation prior to Figure 1.
Please, insert a paragraph on the spatial reference levels of your analysis and provide definitions of the spatial reference levels. What do you mean by “out-of-city areas”? Do you mean the area outside the administrative borders of the cities or do you mean the city plus suburban areas? Please, explain.
Please, provide definitions on the hospital levels.
Results:
This chapter requires special attention:
In order to optimize readability, please shift Tables 1 to 3 (inclusive of captions) to the appendix, the same applies for Figures 5 to 7 (inclusive of captions). Please, provide a textual synthesis of the results of Tables 1 to 3 as well as of Figure 5 to 7 in the “Results” section.
Discussion:
In my point of view, the first paragraph is redundant.
Line 409: referring to SDG 3 “Good Health and Well-Being” than to “the Sustainable Development Goals” would be more appropriate.
Last paragraph: Please, explain the relevance of the “concurrent health insurance payment policy” on the downward trend [of hospital visits] at the end of each year”.
Conclusions:
This section requires specific attention related to wording and content.
As I understand it, the study relates to people who are living in a Chinese province which consists of cities and urban municipalities/counties. Please, clarify.
Moreover, this section mainly presents results (line 434 to line 438).
Please, provide more accurate conclusions supported by your findings.
All the best!
Reviewer 2 Report
The authors make a great effort to analyze all the data presented.
COVID-19 has hit national healthcare systems leading to significant re-organization and redistribution of resources. In addition, COVID-19 highlighted all the critical issues in our healthcare system.
The paper is well written. Methods are precise and results are well documented. The introduction is a bit long, I would advise the authors to reduce the length of the introduction section.
Reviewer 3 Report
Thank you for the opportunity to review this manuscript, "Long-term Impact of COVID-19 on Hospital Visits for Rural Residents in Guangdong, China: A Controlled Interrupted Time Series Study." The manuscript addresses an interesting topic of public health importance in China. The research methods, procedures, and steps have been done logically. The results are interesting and valuable. Some improvements may further strengthen this paper:
- The introduction section is well written, but the first paragraph could be shortened to be easier to read and more focused on the aim of the paper. Also, it would be better if the authors mentioned how the results of this study are expected to help this population.
- In the discussion section, an explanation of how the findings may improve the well-being of this population and their families in long-term care will further strengthen the contribution of the study.
- Strengths and practical implications of this paper and/or describe what needs to be advanced as a prospective future study based on the findings and limitations of the current study.
- Institutional Review Board Statement part of this study should be motioned.
